# Towards Personalized Medicine in Myasthenia Gravis: Role of Circulating microRNAs miR-30e-5p, miR-150-5p and miR-21-5p

**DOI:** 10.3390/cells11040740

**Published:** 2022-02-20

**Authors:** Francesca Beretta, Yu-Fang Huang, Anna Rostedt Punga

**Affiliations:** 1School of Medicine and Surgery, University of Milano-Bicocca, 20900 Monza, Italy; f.beretta30@campus.unimib.it; 2Department of Medical Sciences, Clinical Neurophysiology, Uppsala University, 75185 Uppsala, Sweden; yu-fang.huang@neuro.uu.se

**Keywords:** myasthenia gravis, circulating miRNAs, miR-150-5p, miR21-5p, miR-30e-5p, personalized medicine, biomarker

## Abstract

Myasthenia gravis (MG) is an autoimmune neuromuscular disease characterized by fatigable skeletal muscle weakness with a fluctuating unpredictable course. One main concern in MG is the lack of objective biomarkers to guide individualized treatment decisions. Specific circulating serum microRNAs (miRNAs) miR-30e-5p, miR-150-5p and miR-21-5p levels have been shown to correlate with clinical course in specific MG patient subgroups. The aim of our study was to better characterize these miRNAs, regardless of the MG subgroup, at an early stage from diagnosis and determine their sensitivity and specificity for MG diagnosis, as well as their predictive power for disease relapse. Serum levels of these miRNAs in 27 newly diagnosed MG patients were compared with 245 healthy individuals and 20 patients with non-MG neuroimmune diseases. Levels of miR-30e-5p and miR-150-5p significantly differed between MG patients and healthy controls; however, no difference was seen compared with patients affected by other neuroimmune diseases. High levels of miR-30e-5p predicted MG relapse (*p* = 0.049) with a hazard ratio of 2.81. In summary, miR-150-5p is highly sensitive but has low specificity for MG, while miR-30e-5p has the greatest potential as a predictive biomarker for the disease course in MG, regardless of subgroup.

## 1. Introduction

Myasthenia Gravis (MG) is a chronic autoimmune neuromuscular disorder caused by impaired transmission due to antibodies (Abs) against postsynaptic receptors at the neuromuscular junction. The cardinal symptoms in MG are fatigable skeletal muscle weakness, with fluctuations in symptoms from day to day or even from hour to hour. Based on affected muscle groups ocular MG (OMG) is localized to the extraocular muscles, and generalized MG (GMG) has a general muscle involvement, including bulbar, axial or limb muscles, or even the diaphragm [1]. MG is to date one of the best characterized autoimmune and neurological diseases, although important aspects, such as pathogenesis and biomarkers able to predict the disease course and response to treatment are ongoing areas of research.

MG diagnosis is made on the combination of clinical presentation, and either positive Ab serology or abnormal electrophysiological testing, consisting of either repetitive nerve stimulation (RNS) or single fiber electromyography (SFEMG) [1]. Different MG subgroups have been proposed based on the age of onset [early onset MG (EOMG; onset ≤ 50 years of age); late onset MG (LOMG; onset > 50 years of age)], antibody profile [acetylcholine receptor antibody seropositive (AChR+), muscle specific tyrosine kinase antibody seropositive (MuSK+)], weakness distribution (OMG vs. GMG) and thymic abnormalities [1,2]. Different subgroups reflect important implications in both prognosis and therapeutic management [1,3,4].

Most patients present with an OMG phenotype but progress to a generalized disease within 2 years of diagnosis, and approximately 17% of MG patients remain with an isolated ocular phenotype [5]. Additionally, maximal MG severity in GMG is usually reached within two years of disease onset [5].

Dysregulation of immune mechanisms with loss of tolerance toward key muscle antigens is the hypothesized pathogenic mechanism in MG, although any specific trigger, as well as the exact mechanisms, are still unknown [6]. There are several therapeutic options in MG, which range from symptomatic treatment to disease modifying drugs, rescue therapies and even thymectomy, and clinical remission or good symptom control is achieved in most patients [7]. However, no adequate parameter to predict disease course at onset or response to treatment has been validated to date [8], and therapeutic decisions still depend heavily on the single clinician’s expertise. In the context of randomized clinical trials, several clinical scales have been validated throughout the years, employing objective, patient reported, or composite measures of disease severity [9,10], but are intrinsically limited by their lack of prognostic significance and low sensitivity in differentiating mildly from moderately affected patients. Therefore, reliable biomarkers are urgently needed, both in MG overall, as well as in the different subgroups in clinical practice and clinical trials to evaluate, guide and monitor therapeutic choices [11].

Circulating microRNAs (miRNAs) have emerged as potential biomarkers in MG [8,12]. Specific serum miRNA profiles have been associated with different MG subgroups and correlate with clinical response upon treatment and thymectomy [13,14]. In particular, levels of miR-150-5p and miR-21-5p have been found to be elevated in EOMG and LOMG patients compared to healthy subjects in several studies [13,15,16]. Moreover, miR-150-5p has been shown to correlate with response upon thymectomy in EOMG patients [14], while another study showed how serum exosomal miR-150-5p decreased in parallel with clinical improvement and steroid requirement after treatment with rituximab in AChR+ MG patients [17]. A third miRNA, miR-30e-5p is elevated in LOMG patients’ sera compared to healthy controls and seems to correlate with the risk of generalization in OMG patients [13,18]. In contrast, miR-30e-5p was reduced in a cohort of EOMG AChR+ female patients [15].

The aim of the present study was to determine the sensitivity and specificity of serum miRNAs miR-150-5p, miR-21-5p and miR-30e-5p as biomarkers in all MG patients subgroups at an early disease stage and to evaluate their respective predictive and prognostic potential.

## 2. Materials and Methods

### 2.1. Patient Cohorts

Patient sera were obtained from the Uppsala University biobank or from newly recruited patients at the Department of Clinical Neurophysiology, Uppsala University Hospital, according to the following criteria: MG diagnosis based on typical skeletal muscle fatiguability with at least one confirmatory testing, i.e., a positive AChR or MuSK Ab test and/or abnormal electrophysiology by either RNS or SFEMG; serum had to be sampled ≤ 1 year of diagnosis; with or without ongoing immunosuppressive treatment. Of the 32 MG patients fulfilling the stipulated inclusion criteria, three patients were excluded because of lack of serum sample in the biobank and two more patients were newly recruited for the study.

A control population was selected based on two cohorts. The first group included healthy individuals [healthy controls (HC)] recruited on a volunteer basis from adult blood donors of all ages presenting to Uppsala University Hospital Blood Central. Serum samples from 46 male and 49 female HCs were already present in the Biobank (collected in 2014), while a total of 200 serum samples (100 males and 100 females) were collected from March to May 2021. Therefore, a total of 295 HCs (146 males, 149 females) were included. The second cohort included 24 patients with non-MG autoimmune neurological disorders without ongoing immunosuppressive treatment from the Neurology Biobank at Uppsala University Hospital. This group of other neuroimmune disorders (OND) included patients with multiple sclerosis (n = 15), Lambert–Eaton myasthenic syndrome (n = 2), Guillain Barré syndrome (n = 2), chronic inflammatory demyelinating polyneuropathy (n = 4), and inflammatory myelitis (n = 1).

Clinical records of all MG patients were reviewed to assess gender, age at MG onset, timing from symptoms onset to diagnosis, clinical subgroup and severity, serological subgroup, electrophysiological data (RNS/SFEMG), ongoing and previous treatments, and disease course (stability vs. flares) at follow-up. All patients were diagnosed and evaluated by physicians who are experts in the field of MG. Clinical subgroups were identified as EOMG or LOMG and OMG or GMG, and disease severity assessed though validated scales: Myasthenia Gravis Foundation of America (MGFA) clinical classification and the MG composite score (MGC).

This study was approved by the Swedish Ethical Review Authority on human experimentation (ethical permit number 2020-03049) and written informed consent for research was obtained from all patients and controls.

### 2.2. Blood Samples, RNA Isolation and qPCR

Blood samples were collected in tubes without any additives, stored at room temperature for at least 20 min and then centrifuged. The centrifugation protocol was 1200× *g* for 5 min for samples collected before 2021, and 2200× *g* for 10 min for samples collected after January 2021 because of a change in vial supply. All samples were stored at −80 °C until further processing. After thawing on ice, serum samples were centrifuged at 1500× *g* for 5 min at 4 °C. Total circulating RNA was isolated from 200 µL of serum with the miRCURY^®^ RNA Isolation Kit-Biofluids (Exiqon^®^ #300112, Vedbaek, Denmark) according to the manufacturer’s instructions for samples before 2021. Because of the change in production, isolation of RNA after 2021 was performed using the miRNeasy^®^ Serum/Plasma Advanced kit (QIAGEN^®^ #217204, Venlo, The Netherlands). To align the reaction to the previously used protocol, total RNA was eluted with 50 µL of nuclease free water in the final step. Synthetic RNA spike-ins UniSp2, UniSp4 and UniSp5 (RNA Spike-in kit for RT, QIAGEN^®^ #339390, Venlo, Netherlands) were added to lysis buffer to monitor RNA isolation efficiency. To improve RNA extraction yield, 1 µg of MS2 bacteriophage RNA (Roche^®^, #10165948001, Basel, Switzerland) was also added. All samples were stored at −80 °C until further processing.

Isolated RNA (2 µL) was used for cDNA synthesis through reverse transcription (RT) in 10 µL reaction mix using the miRCURY^®^ LNA^®^ RT kit (QIAGEN ^®^ #339340, Venlo, The Netherlands) according to the manufacturer’s instructions. UniSp6 was added at this stage to check for RT and polymerase chain reaction (PCR) inhibitors. All cDNA was newly synthesized and then stored at −20 °C for up to 24 h before performing real-time quantitative PCR. Real-time quantitative PCR (qPCR) was used to quantify miR-30e-5p expression, employing the miRCURY^®^ LNA^®^ SYBR^®^ Green PCR kit (QIAGEN^®^ #339347 and #339346, Venlo, The Netherlands) according to the manufacturer protocol. Custom RT-qPCR panels (miRCURY^®^ LNA^®^ miRNA custom PCR Panel, QIAGEN^®^, Venlo, The Netherlands) were designed to contain primers for our target miRNAs, endogenous miRNAs for expression analysis, quality control and hemolysis controls, and synthetic RNA spike-ins for quality control and final calibration. A final cDNA dilution (50×) was used for each 10 µL reaction mix. Two replicas of each target miRNA and candidate reference miRNA were performed. See Table A1 (Appendix A) for specific miRNAs’ sequences and assays used. Amplification was performed with the ABI^®^ QuantStudio^®^ 6 Flex Real-Time PCR System (Thermo Fisher Scientific^®^, Waltham, MA, USA). Amplification curves and melting curve analysis were performed to check for correct amplification through the QuantStudio^®^ Real-Time PCR software (Thermo Fisher Scientific^®^, Waltham, MA, USA), samples with a ΔCq of replicas > 1.5 or Ct > 36 were excluded. A ΔCq of replicas > 1.5 was chosen as the cut-off to avoid the exclusion of samples with low miRNA concentration (high Cq), as increasing Cq variance is expected in these cases [19,20]. Majority of replicas (~97.2%) had a ΔCq < 0.5. Further quality control (QC) was assessed with the use of synthetic RNA spike-ins UniSp2, UniSp4, UniSp5, UniSp6 and UniSp3, and evaluation of endogenous miRNA expression as suggested by Qiagen^®^. Interplate calibration was achieved with UniSp3. To exclude cellular miRNA contamination, all samples were checked for possible hemolysis using ΔCq (miR-23a-3p–miR-451a) [21].

Comparative Ct method was used to quantify miR-30e-5p, miR-150-5p and miR-21-5p expression using the ΔCq method and the following formula [22]:target miRNA expression relative to reference miRNA = 2^−ΔCq^,(1)
where ΔCq = average Cq miRNA of interest–average Cq reference miRNA.

Reference candidates in this study were miR-191-3p and miR-103a-3p. Final normalization was performed to miR-191-3p as it was more consistently expressed, in line with previous studies [15,18]. Log_2_ conversion was performed to normalize the data before proceeding to statistical analysis, according to the following formula:miR-30e-5p (relative to miR-191-3p) = Log2 (2^−ΔCq^ ∗ 100).(2)

### 2.3. Study Design and Statistical Analyses

Levels of miRNA were presented as mean ± standard deviation (SD), while non-parametric data (follow-up times, etc.) were presented as median with 1st and 3rd interquartile ranges (IQR).

We first compared miR-30e-5p between EOMG and LOMG using an unpaired two-tailed t-test, as different expression values have been described in these subgroups compared to controls [13,15]. If no statistically significant difference was found, MG patients were to be grouped together for all subsequent analyses to increase power. Analysis of variance (one-way ANOVA) was performed for all miRNAs, the null hypothesis being that the mean values for the target miRNAs were the same across the three groups (MG, OND and HC). If a statistically significant difference was found, post-hoc analyses with Tukey’s multiple comparisons test were performed to obtain adjusted *p* values. A *p* < 0.05 was considered statistically significant. To analyze miRNAs in MG as diagnostic tests, receiver operating characteristic (ROC) curves were plotted using the Wilson-Brown method to determine the area under the curve (AUC) with respective 95% confidence intervals (CI) and associated *p*-value. Youden’s index was used to determine sensitivity and specificity at the best trade-off.

The second part of the study was designed as a “survival analysis” to see whether miRNAs expression values could predict clinical changes in MG patients. T0 was defined as the time at diagnosis and the endpoint as the time to a meaningful disease worsening (flare) in months. Disease status was based on MGFA clinical classification, with generalization defined as a change from MGFA class 1 to a class ≥ 2 and meaningful disease relapse defined as a change in MGC of > 3 points or change from a lower to a higher MGFA class. Patients were divided into two groups based on miRNAs serum levels, group 1 “high miRNAs” and group 2 “low miRNAs”, with the cut-off derived from the analyses performed on our own dataset. The respective Kaplan–Meyer survival curves were drawn, and the Log-rank (Mantel–Cox) test was used to compare the two curves. Hazard ratio (Mantel–Haenszel) with 95% CI was also calculated.

Statistical analysis and graphs were made using GraphPad Prism^®^ (GraphPad Software Inc., San Diego, CA, USA).

## 3. Results

### 3.1. Patients and Controls Demographics

The final MG cohort consisted of 31 patients, 10 EOMG and 21 LOMG; 295 HCs (146 males, 149 females) and 24 patients with ONDs were included via sample collections and the biobanks. After qPCR analysis, 48 samples were excluded because of incorrect amplification (2 MG patients, 42 HCs, 4 ONDs) or a hemolysis score ≥ 7 (2 MG patients and 8 HCs). A total of 292 patients, whose serum samples passed quality controls, were included in the final expression analyses: 17 LOMG, 10 EOMG, 245 HCs, 20 ONDs.

Table 1 summarizes the demographics and clinical characteristics of the MG cohort. Patients included had been followed up for a median of 4 years (IQR: 1 year; 8 years).

Of the 245 HCs included in the final analysis, 129 (53%) were females and their age ranged from 20 years to 79 years, with a median of 44 years (IQR: 31 years; 55 years). In the OND group of 20 patients [13 (65%) females], the median age was 46 years (IQR: 33.3 years; 59.3 years) and all were immunosuppressive naïve. The diseases subgroups represented in the OND group during the final analysis were: 13 patients with multiple sclerosis (MS, three males and 10 females), two patients with Lambert–Eaton myasthenic syndrome (LEMS, one male and one female), four patients with chronic inflammatory demyelinating polyneuropathy (CIDP, two males and two females), and one patient with inflammatory myelitis (IM, one male). In the OND cohort, full information on comorbidities was not available, but the OND patients did not have any other specified autoimmune diseases and they were all immunosuppressive naïve at sampling.

### 3.2. Serum miRNA Levels

#### 3.2.1. Primary Analyses

Since no significant difference in serum miR-30e-5p levels was found between EOMG and LOMG patients (6.87 ± 1.03 and 6.68 ± 0.78, respectively; *p* = 0.59), all MG patients were grouped together in subsequent analyses. Comparison among the three cohorts (MG, OND and HC) indicated significant difference for miR-30e-5p (*p* < 0.0001), miR-150-5p (*p* = 0.0009) and miR-21-5p (*p* = 0.0002). Post-hoc analyses confirmed a significant difference between MG patients and HCs for miR-30e-5p (6.75 ± 0.87 and 6.06 ± 1.01, respectively; *p* = 0.0016) and miR-150-5p (6.93 ± 1.07 and 6.16 ± 1.10, respectively; *p* = 0.0015). OND patients had higher levels than HCs for miR-30e-5p (6.97 ± 0.60 and 6.06 ± 1.01, respectively; *p* = 0.0002) and miR-21-5p (9.97 ± 0.72 and 9.27 ± 0.77, respectively; *p* = 0.0004). Nevertheless, miR-150-5p levels were comparable between OND and HC (6.58 ± 0.80 and 6.16 ± 1.10, respectively; *p* = 0.21). Levels of miR-21-5p did not significantly differ between MG and HCs (9.58 ± 0.87 and 9.27 ± 0.77, respectively; *p* = 0.13). Levels between MG and OND patients for any miRNAs also did not differ significantly [miR-30e-5p (6.75 ± 0.87 and 6.97 ± 0.60, respectively; *p* = 0.74); miR-150-5p (6.93 ± 1.07 and 6.58 ± 0.80, respectively; *p* = 0.52); miR-21-5p (9.58 ± 0.87 and 9.97 ± 0.72, respectively; *p* = 0.20)]. Expression ranges for all miRNAs using box plots are represented in Figure 1.

We noticed a wider range of expression than expected in the HC population. Post-hoc comparison between sera of HCs collected in 2014 and ones collected in 2021 showed a significant difference in both miR-150-5p and miR-21-5p levels between these two groups (*p* < 0.0001 for both miRNAs). Remarkably, comparison between the “old controls” and MG patients for miR-21-5p showed significant results (EOMG vs. old controls *p* < 0.0001, LOMG vs. old controls *p* = 0.0187). These data are also represented in Figure 1.

#### 3.2.2. Secondary Analyses

In the MG cohort, no correlation was found between any of the three miRNAs and other clinical parameters, such as age, sex, MGFA class, MGC score, or Ab class. Further, no correlation was found between miRNA levels and age or sex in the HC cohort.

To evaluate whether immunosuppressive treatment (30% of our cohort) might have influenced our results, we compared miRNA levels between immunosuppressive MG naïve patients with those with ongoing treatment (two-tailed unpaired t-test assuming unequal variances). There was no statistically significant difference between the two groups regarding all miRNAs (miR-30e-5p *p* = 0.98, n.s.; miR-150-5p *p* = 0.27, n.s.; miR-21-5p *p* = 0.58, n.s.).

Several MG patients had comorbidities (Table 1). We broadly divided patients in four categories: patients without comorbidities; MG-related comorbidities, i.e., thymoma; autoimmune diseases, which included vasculitis (n = 1), atopic and allergic diseases (n = 4), type 1 diabetes mellitus (n = 2), and autoimmune hypothyroidism (n = 1); and other comorbidities, consisting of age-related cardiovascular disorders (n = 8), neoplasia other than thymoma (n = 1), Parkinson’s disease (n = 1). When comparing miRNA values among these four groups (ANOVA), no difference was found between any specific subgroup (data not shown). 

ROC curves for miR-30e-5p and miR-150-5p were plotted for both MG and OND patients against HCs (Figure 2). We found an AUC of 0.69 (95% CI 0.58–0.80; *p* = 0.0013) for miR-30e-5p in MG patients, with a sensitivity of 55.6% (95% CI 37.31–72.41%) and a specificity of 85.7% (95% CI 80.8–89.6%) at best trade-off (Youden’s index 0.4). The value of miR-30e-5p at best trade-off was 6.89. AUC for OND patients was 0.80 (95% CI 0.71–0.89; *p* < 0.0001). At best trade-off (Youden’s index 0.5), sensitivity was 70% (95% CI 48.1–85.5%) and specificity was 81.6% (95% CI 76.3–86.0%). In summary, miR-30e-5p did not perform well as a diagnostic test for MG.

ROC curves for miR-150-5p showed an AUC of 0.70 (95% CI 0.61–0.79; *p* = 0.0008) with sensitivity of 85.2% (95% CI 67.5–94.1%) and specificity of 48.2% (95% CI 42.0–54.4%) at best trade-off for MG patients (Youden’s index 0.3), and an AUC of 0.63 (95% CI 0.52–0.73; *p* = 0.06, n.s.) with sensitivity of 80.0% (95% CI 58.4–91.9%) and specificity of 52.2% (95% CI 46.0–58.4%) at best trade-off for OND patients (Youden’s index 0.3). We decided to perform the analysis also for the EOMG subgroup, as there was a slightly different distribution, albeit not significant, in this subgroup. We found an AUC of 0.77 for EOMG (95% CI 0.65–0.89; *p* = 0.004) with sensitivity and specificity of 90% (95% CI 59.6–99.5%) and 58.4% (95% CI 52.11–64.4%), respectively, at best trade-off (Youden’s index 0.5). As such miR-150-5p performs best in EOMG, with high sensitivity but still lacking in specificity. Additionally, miR-150-5p did not perform well as a diagnostic test.

#### 3.2.3. Predictive Value of miR-30e-5p in MG Progression

We decided to use the level of miR-30e-5p that we obtained at best trade-off from the ROC curves, i.e., 6.89, to divide our MG population into two groups: group 1 “high miR-30e-5p” (levels ≥ 6.89) included 15 patients (six EOMG, nine LOMG); group 2 “low miR-30e-5p” (levels < 6.89) included 12 patients (four EOMG, eight LOMG). Kaplan–Meyer survival curves based on time to disease relapse in months separated these two groups (Figure 3). During follow-up, a total of 15 patients (six EOMG and nine LOMG) underwent worsening (flare) defined as a change in MGC of > 3 points or passage from a lower to a higher MGFA class. The median time to worsening was 7 months (IQR: 3.5–13 months). miR-30e-5p levels correlated with disease flare (*p* = 0.0495). The hazard ratio was 2.81 (95% CI 1.00–7.88).

In our patient cohort, there were 10 OMG patients, the majority of which generalized during follow-up. As such, it was deemed meaningless to perform additional testing using generalization as endpoint and data were approached descriptively. The majority of ocular patients who generalized (three out of four EOMG and all four LOMG patients) belonged to the “high miR-30e-5p” group (EOMG: 7.29 ± 0.78; LOMG 7.63 ± 0.26).

Because of the high sensitivity for MG displayed by miR-150-5p, a specific cut-off could not be established to subdivide our MG cohort into high and low expression subgroups, and additional survival analyses were not performed using this miRNA.

## 4. Discussion

We analyzed serum levels of miR-150-5p, miR-21-5p and miR-30e-5p in MG patients belonging to different subgroups in order to further elucidate their role as biomarkers in MG, trying to understand how they could be best employed in the clinical management of patients. Our data indicate that both miR-150-5p and miR-30e-5p expression levels are significantly higher in all MG patients compared to HCs, in line with previous studies [8]. Nevertheless, miR-21-5p levels did not significantly differ between MG patients and a large number of HCs. The diagnostic utility of both miR-150-5p and miR-30e-5p is limited, given the important overlap in levels between MG patients, HCs and OND patients. This finding is not surprising since miRNAs are key mediators of gene expression, and their alteration could be either causative or a consequence of the disease process itself. Both miR-150-5p and miR-21-5p have been labeled “immuno-miRs,” a subset of miRNAs involved in the regulation and modulation of immune cell functions [23]. miR-150-5p has been shown to be involved in T cell maturation and is an important regulator of both natural killer (NK) and B cells [8]. Moreover, it is proposed as a modulator of both CD4+ and CD8+ T cells survival and is strongly expressed in B cells in the thymus and particularly in those surrounding germinal centers [24], possibly sustaining an ongoing immune response in MG. miR-21-5p has also been proposed to regulate several processes important for T cell activation and has been investigated as an anti-apoptotic agent, especially in cancer, and has been found to be increasingly expressed in regulatory T (Treg) cells [23]. The last miRNA, miR-30e-5p, has been linked to a variety of diseases, mainly cancer, in which it has been proposed both as a protective and favoring factor depending on the underlying mechanism [25,26].

All three miRNAs are linked with several proinflammatory pathways [27] and are altered in other autoimmune diseases, including neurological disorders. Upregulation of nuclear factor kappa-light-chain-enhancer of activated B cells (NF-kB) in a murine macrophage cell line caused a statistically significant increase in intracellular levels of both miR-30e-5p and miR-21-5p and promoted their packaging into exosomes, possibly explaining the finding of elevated serum levels in MG patients [27]. Keller et al. showed that miR-30e-5p is increasingly expressed in peripheral blood mononuclear cells (PBMCs) of MS patients compared to HCs [28], and increased plasma levels of this miRNA were detected in relapsing-remitting MS patients compared to HCs [29]. This study also proposed interleukin 10 (IL-10), a key immunoregulatory cytokine, as a possible target of miR-30e-5p [29]. Another autoimmune disease in which miR-30e-5p has been shown to be altered is systemic lupus erythematosus (SLE) [30,31]. Regarding miR-150-5p, there are several reports of its dysregulation in Sjögren Syndrome [32,33], in which it accumulates in salivary glands in a manner comparable to the thymus in MG. Another study showed downregulation of this miRNA in PBMCs in Sjögren Syndrome and upregulation, albeit not significant, in SLE compared to controls [34]. Moreover, miR-150-5p levels were elevated in cerebrospinal fluid of MS patients [35], and its suppression was shown to reduce disease severity in a murine model of autoimmune encephalitis [36].

Serum extracellular vesicle miR-21-5p was found to be significantly elevated in patients with type 1 autoimmune pancreatitis and sera of patients with psoriatic arthritis [37,38]. Again, also this miRNA has been found dysregulated in MS, in particular, values have been shown to be increased in PBMCs of patients in remission compared with patients undergoing disease relapse [39]. Overlap between MG and OND patients is, therefore, not surprising, and the significantly different expression between OND patients and HCs suggests that these miRNAs are involved in the underlying neuroimmune process. Further studies are needed to understand the role of miRNAs in the pathogenesis of both MG and other neuroimmune diseases, as understanding pathophysiology is key to developing targeted therapeutic options.

We did not find differences in miRNA expression when comparing MG patients undergoing immunosuppression with treatment naïve patients, although it is possible that the influence of immunosuppressive treatment is modest and was masked by other confounding factors in our small sample. Ongoing immunosuppressive treatment could, however, explain the finding of overall higher expression of both miR-30e-5p and miR-21-5p in the OND group, who were all immunosuppressive naïve, compared to MG patients. It is, however, intriguing how overall values of miR-150-5p, and especially the ones in EOMG, were higher compared to ONDs regardless of immunosuppressive status. We also did not find a specific correlation between miRNA levels and either age or sex in the large HC cohort. Therefore, despite HC and OND cohorts being younger and having a female predominance compared to MG patients we exclude the role of age and sex as possible confounding factors.

One puzzling finding in our dataset was the unexpectedly high overlap in miRNA expression between MG patients compared to newly collected HC serum samples. We noticed a marked difference between serum samples of HCs recruited in 2021 compared to ones recruited in 2014, which was significant for both miR-150-5p and miR-21-5p. One of our hypotheses is that the ongoing COVID-19 pandemic, which was in the middle of its 3rd peak in Sweden during serum sample collection, could have influenced the expression of our target miRNAs. As previously mentioned, miR-21-5p and miR-30e-5p levels have been shown to be increased both intracellularly and into exosomes by upregulation of the NF-kB pathway through stimulation with lipopolysaccharide [27], therefore, dysregulation of this pathway provides a theoretical link to innate immunity and to SARS-CoV-2 infection specifically [40]. Moreover, changes in miRNA expression have been proposed to happen due to infectious stimuli [6] and MG as well as the diseases included in the OND group, excluding LEMS, which is a well described paraneoplastic syndrome, have all been postulated to possibly originate from a remote viral infection [6,41,42]. To exclude an influence of the ongoing pandemic on our results, it would be interesting to measure specific markers related to SARS-CoV-2 in our healthy controls’ serum samples, especially those displaying very elevated miRNA values. Other unforeseen conditions could also potentially influence these miRNA levels. We assumed that volunteer blood donors are healthy individuals who frequently undergo screening tests. Nevertheless, we do not have access to any additional medical information on these subjects, especially concerning yet to be diagnosed or “mild” autoimmune disorders, such as Hashimoto’s thyroiditis, which would not prevent blood donations. Another explanation that must be taken into consideration is that of analytical or technical differences in sample handling, including prolonged storage. We are inclined to exclude this hypothesis since both endogenous control miRNAs, miR-191-3p and miR-103a-3p, and synthetic RNA spike-ins, added in RNA samples before storage in the biobank, were consistently well expressed in samples that passed quality control, and the difference between controls was remarkable in only two of our target miRNAs. Changes in blood vials, centrifugation protocol for serum collection and different RNA extraction kits were possibly important technical differences between stored and newly collected samples but are insufficient to explain the statistically significant difference we have found.

The range of miR-30e-5p expression levels we obtained in this study was lower compared to the previous one by Sabre et al. [18]; therefore, it was not possible to apply the previously proposed cut-off of 8. This could be explained by the method itself, as what we obtain is a relative value, representing the fold expression change of the target miRNA compared to another stable endogenous miRNA [22]. An important finding in our study is how higher miR-30e-5p values correlate with increased disease activity, as it was shown that MG patients with serum levels ≥ 6.89 were more susceptible to undergo disease relapse compared to patients with a lower miR-30e-5p value. At the same time, miR-30e-5p did not correlate with other clinical parameters, especially MGC and MGFA class, when considered at a given time-point. In our opinion, this implies that miR-30e-5p correlates with disease progression regardless of clinical severity status. miR-30e-5p levels have already been proposed as a risk factor for generalization [18], and clinical course in LOMG patients [13]. In our cohort, seven out of eight patients who generalized were in the “high miR-30e-5p group”, and, although limited in number, we believe that also our findings support a correlation between miR-30e-5p expression levels and generalization. Qualities displayed by miR-30e-5p in our and previous studies, thus, suggest a role as a prognostic or predictive biomarker in MG patients regardless of disease subgroup. Furthermore, we believe that the elevated miR-30e-5p levels we found also in EOMG patients are also in support of this hypothesis.

This study has limitations. The study design was in large retrospective, which is an important limit for the survival analysis. MGC scores were not available for all patients, and this was the reason why we used a combination of MGFA class and MGC score as the endpoint, even though MGFA class as a qualitative estimate may not be an optimal endpoint to evaluate disease activity [43]. Further, some patients had a relatively short follow-up compared to others, and although censored data are accounted for in the statistical method, they must be kept in mind during analysis interpretation. Moreover, the survival analysis was underpowered. Post-hoc power analysis performed using the table proposed by Freedman [44] revealed that the needed number of patients to reach a power of 0.8 with a significance of 0.05 for a follow-up period of 2 years (24 months), would be of 42 for each group, and we probably reached significance thanks to the very long observation period in some patients (about 10 years). From previous epidemiological studies, we know that most MG patients worsen, whether generalizing from an ocular onset or reaching a more severe disease state, within the first two years of disease onset which is partially reflected by our data [5]. Additionally, looking at our survival curves we could observe a divergence between the two groups as early as after 10 months. We believe that our findings in respect to miR-30e-5p warrant a correctly powered follow-up study with a prospective design. MG is a rare disorder, and if we consider its incidence rate, the involvement of multiple centers would be advisable to reach the needed number of patients in a reasonable timespan. A combination of putative biomarkers, including different miRNAs or a combination of miRNAs and proteins, could possibly better reflect the risk profile of MG patients. Therefore, other biomarkers should also be analyzed together with MG associated miRNAs in any following studies, so that targeted and personalized treatment tailored to the single patient could finally be realized.

## 5. Conclusions

We propose miR-30e-5p as a predictive biomarker in MG, as its levels correlate with disease course but not with severity itself, while miR-150-5p appears once again to be the most MG-sensitive miRNA, especially in the EOMG subgroup. However, none of the miRNAs we analyzed seem to be strictly MG specific since they are found elevated also in other neuroimmune diseases. A correctly powered, multicenter, longitudinal prospective study is needed to confirm and validate our data regarding miR-30e-5p. Lastly, technical improvements are also needed to translate the use of these biomarkers from the research setting to the clinic.

## Figures and Tables

**Figure 1 cells-11-00740-f001:**
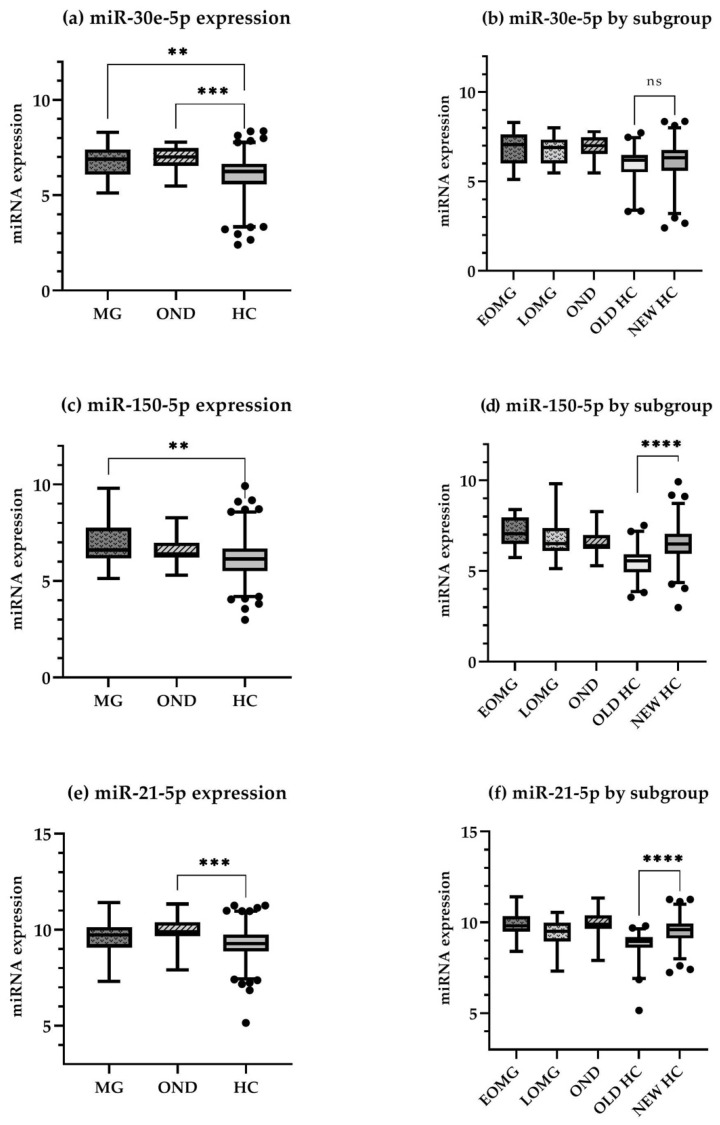
Box plots (2.5–97.5 percentile, outlying values are marked as black dots) representing expression ranges for all miRNAs in different cohorts, statistically significant differences are marked with asterisks (ns *p* > 0.05, ** *p* ≤ 0.01, *** *p* ≤ 0.001, **** *p* ≤ 0.0001). (**a**,**b**) miR-30e-5p, (**c**,**d**) miR-150-5p, (**e**,**f**) miR-21-5p. In the graphs depicting miRNAs expression levels by subgroup, a significant difference in the distribution has been observed between old and new healthy controls for miR-150-5p and miR-21-5p. Additionally, while not statistically significant, a slight difference in miR-150-5p and miR-21-5p expression between EOMG and LOMG can be observed, with a tendency of higher values in the EOMG subgroup. Values represent Log_2_ converted data. Abbreviations: HC, healthy controls; MG, myasthenia gravis; OND, other neuroimmune diseases; EOMG, early onset MG; LOMG, late onset MG.

**Figure 2 cells-11-00740-f002:**
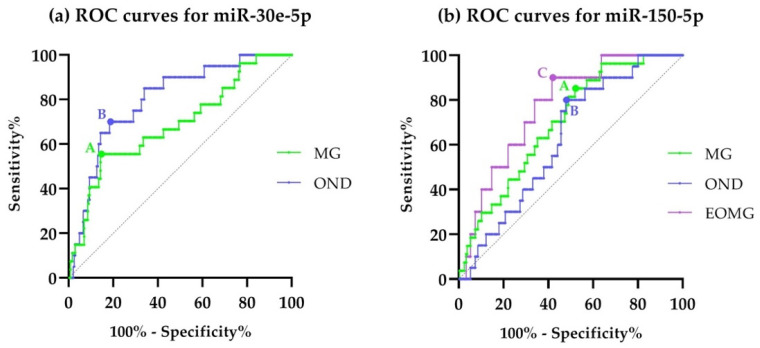
(**a**) ROC curves for miR-30e-5p of cases against controls. The green curve represents MG patients, while the blue curve shows OND patients. The best trade-off for MG patients is represented by point A while the best trade-off for OND is at point B. Both curves are statistically significant but suboptimal; (**b**) ROC curves for miR-150-5p, same colors and letters apply for MG and OND patients, in violet curves for EOMG with C representing the point at best trade-off.

**Figure 3 cells-11-00740-f003:**
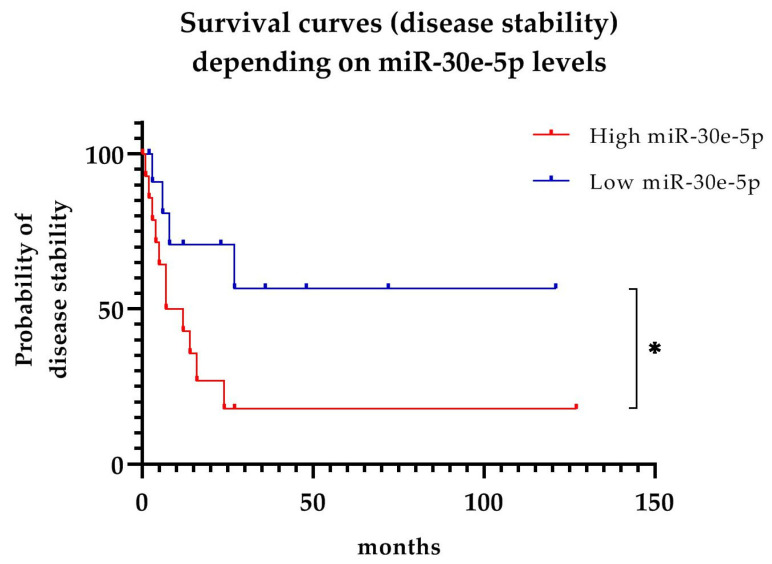
Survival curves (disease stability) for MG patients depending on serum levels of miR-30e-5p. A gap can be seen between the two curves as early as 10 months from onset, and a statistically significant difference (*p* = 0.049) is present. Bars represent censored data. * *p* ≤ 0.05.

**Table 1 cells-11-00740-t001:** Demographics and clinical characteristics of MG patients.

	All MG	EOMG	LOMG
Total patients	27	10	17
Sex			
F	15 (55.5%)	8 (80%)	7 (41%)
M	12 (44.5%)	2 (20%)	10 (59%)
Age (y) [median (IQR)]	58 (39; 69.5)	32.5 (25; 41.5)	68 (52; 75)
Time from diagnosis (months; mean ± SD)	3.9 ± 4.7	5 ± 4.7	3.4 ± 4.8
Serology:			
AChR+	17 (63%)	4 (40%)	13 (76%)
MuSK+	3 (11%)	2 (20%)	1 (6%)
AChR/MuSK seronegative	7 (30%)	4 (40%)	3 (18%)
Comorbidities:			
none	6 (22%)	3 (30%)	3 (18%)
thymoma	3 (11%)	2 (20%)	1 (6%)
autoimmune	8 (30%)	3 (30%)	5 (29%)
other	10 (37%)	2 (20%)	8 (47%)
OMG at diagnosis	10 (37%)	4 (40%)	6 (35%)
GMG at diagnosis	17 (67%)	6 (60%)	11 (65%)
Immunosuppressive naïve (at testing)	19 (70%)	7 (70%)	12 (71%)
Thymectomy	9 (33%)	5 (50%)	4 (23,5%)
Generalized during FU	8 (30%)	4 (100%)	4 (67%)
Time to generalization (m) [median (IQR)]	6 (3.75; 7.25)	7 (6.25; 7.25)	4 (2.75; 6.75)
Disease relapse during FU	15 (55.5%)	6 (60%)	9 (53%)
Time to relapse (m) [median (IQR)]	7 (3.5; 13)	7 (6.25; 7.75)	5 (3; 16)

Abbreviations: F, female; M, male; y, years; m, months; FU, follow up. Other comorbidities include non-autoimmune chronic disorders, for example cardiovascular disease.

## Data Availability

The datasets analyzed during the present study are available from the corresponding author on reasonable request.

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
