# Peer review of "Towards Personalized Medicine in Myasthenia Gravis: Role of Circulating microRNAs miR-30e-5p, miR-150-5p and miR-21-5p"

_cells, 2022, doi:10.3390/cells11040740_

Round 1
Reviewer 1 Report
In this study by Beretta et al, several miRNA´s are studied in patients with myasthenia gravis (MG) compared with healthy controls (HC) and patients with other autoimmune neurological disorders including MS GBS etc. The authors report higher levels of two miRNA´s (miR-30e-5p and miR-150-5p) compared with HC. Furthermore, they report high levels of miR-30e-5p to predict an early relapse of MG.
This study has some new information about miRNA in MG, however, there are some concerns as outlined below.
-
There is no information about any comorbidity in MG patients or neurological controls. This may influence the miRNA levels and confound the findings. Therefore, this information needs to be given, and if possible corrected for.
-
It remains unclear how many MG pts received immunosuppressant or immunomodulatory treatment. There may be an effect on miRNA levels due to these treatments and this needs to be described in more detail.
-
There are only minor differences in miRNA levels comparing the three groups. Although some differences are statistically significant, there is considerable overlap between the groups (Figure 1). This hampers the usefulness of measuring miRNA in MG and therefore the statement in the abstract saying that miR-150-5p is the most sensitive miRNA for MG needs to be rephrased. It is unclear what this means and it is not supported by the findings.
-
As eluded to by the authors the levels of miRNA obtained from two different time periods was different. Could this be due to time dependent degradation or other? Could this also have an influence on the findings in MG patients?
-
In figure 3, the authors demonstrate that high levels of miR-30e-5p is predicative for an imminent relapse. This is based on a dichotomous separation of all values of miR-30e-5p by the authors. It would be interesting to see the correlation between values of miR-30e-5p an time to relapse.
Author Response
Reviewer 1
In this study by Beretta et al, several miRNA´s are studied in patients with myasthenia gravis (MG) compared with healthy controls (HC) and patients with other autoimmune neurological disorders including MS GBS etc. The authors report higher levels of two miRNA´s (miR-30e-5p and miR-150-5p) compared with HC. Furthermore, they report high levels of miR-30e-5p to predict an early relapse of MG.
This study has some new information about miRNA in MG, however, there are some concerns as outlined below.
- There is no information about any comorbidity in MG patients or neurological controls. This may influence the miRNA levels and confound the findings. Therefore, this information needs to be given, and if possible corrected for.
Response: We have added information about comorbidities of the MG cohort in table 1 and in the text on p 6 (lines 257-262). In the OND cohort, full information on comorbidities was not available, but the OND patients did not have any other specified autoimmune diseases and they were all immunosuppressive naïve at sampling. We also added this sentence on p 5 (lines 215-217). When redoing the analysis on miRNA expression in subgroups from table 1 based on comorbidities, we found no significant difference (added to Results p 6, lines 262-265). However, these subdivided cohorts are small, and we cannot completely rule out an influence of eventual comorbidities on miRNA values but this is a general in a disease like MG with frequent comorbidities.
- It remains unclear how many MG pts received immunosuppressant or immunomodulatory treatment. There may be an effect on miRNA levels due to these treatments and this needs to be described in more detail.
Response: We have highlighted the characteristics of the MG cohort in table 1. The majority of patients (70%) had no immunosuppressive treatment at initial analysis. We analyzed miRNA expression between MG patients receiving immunosuppression and immunosuppressive naïve patients and found no difference, which is now stated in the Results on page 6 (lines 254-255). The scope of this paper was however not to detect differences at baseline regarding immunosuppression status, since this has previously been studied more in detail in much larger cohorts both retrospectively and prospectively (Punga, A.R., et al., Disease specific signature of circulating miR-150-5p and miR-21-5p in myasthenia gravis patients. J Neurol Sci, 2015. 356(1-2): p. 90-6. Sabre, L., et al., Circulating microRNA miR-21-5p, miR-150-5p and miR-30e-5p correlate with clinical status in late onset myasthenia gravis. J Neuroimmunol, 2018. 321: p. 164-170.)
- There are only minor differences in miRNA levels comparing the three groups. Although some differences are statistically significant, there is considerable overlap between the groups (Figure 1). This hampers the usefulness of measuring miRNA in MG and therefore the statement in the abstract saying that miR-150-5p is the most sensitive miRNA for MG needs to be rephrased. It is unclear what this means and it is not supported by the findings.
Response: We agree with the reviewer that none of the three miRNAs perform sufficiently well as diagnostic test, especially because they are not MG specific (no overall difference with OND, important overlap with controls). We have rephrased our statement regarding miR-150-5p sensitivity in the abstract that sensitivity was high but specificity low. Our data indicate that miR-150-5p was most sensitive in detecting in particular EOMG in regard to the ROC curves at best trade-off, as all MG patients would fall in the high spectrum of expression; nevertheless, it was still not specific enough to be of diagnostic use.
- As eluded to by the authors the levels of miRNA obtained from two different time periods was different. Could this be due to time dependent degradation or other? Could this also have an influence on the findings in MG patients?
Response: We did take technical factors into consideration but removed this point from discussion after final draft revision. In line with this point raised by the reviewer, we reinserted the following section to the discussion part on page 11 (lines 414-423): “Another explanation which must be taken into consideration is that of analytical or technical differences in sample handling, including prolonged storage. We are inclined to exclude this hypothesis since both endogenous control miRNAs, miR-191-3p and miR-103a-3p, and synthetic RNA spike-ins, added in RNA samples before storage in the biobank, were consistently well expressed in samples which passed quality control, and the difference between controls was remarkable in only two of our target miRNAs. Changes in blood vials, centrifugation protocol for serum collection and different RNA extraction kits were possibly important technical differences between stored and newly collected samples but are insufficient to explain the statistically significant difference we have found.”
In summary, technical aspects, especially storage, cannot explain these findings.
- In figure 3, the authors demonstrate that high levels of miR-30e-5p is predicative for an imminent relapse. This is based on a dichotomous separation of all values of miR-30e-5p by the authors. It would be interesting to see the correlation between values of miR-30e-5p an time to relapse.
Response: This is a valid thought raised by the reviewer. Although we did find a trend towards negative correlation between miR-30e-5p levels and time to relapse, the great dispersion (r 2 = 0.02) does not make it relevant. As the data points are few and it does not add meaningful information we did not include it in the text. We do believe that a prospective study with fixed time points of follow-up would be required to respond to this specific query.

Reviewer 2 Report
General comments
- In this manuscript, the authors evaluated the serum levels of miRNAs in 27 MG patients, 245 healthy individuals and 20 patients with non-MG neuroimmune diseases and found high levels of miR-30e-5p predicted MG relapse (p=0.049) with the greatest potential as a predictive biomarker for the disease course in MG, regardless of subgroup. The paper should not be published in current context because of the criticisms as follows
Major criticisms:
- miRNAs have emerged as essential post-transcriptional regulators of gene expression and been evaluated as disease biomarkers for recent 2 decades. However, there were lots of confounding factors interfering the expression of miRNA including immunosuppressive agents. Eight out of 27 MG patients were NOT immunosuppressive naive, which might directly disturb the homogeneity of their MG cohort. Re-analysis using 19 immunosuppressive naive cohort might provide more solid base of comparison.
- Clinical flares, the key event of disease course in this study, were defined as a change in MGC of > 3 points or change from a lower to a higher MGFA class and were documented in 15 out of 27 MG patients. Nerveless, the authors claimed the MGC scores were not available for all patients, and MGFA classes were not an optimal end point for clinical use. In addition, the treatment modalities including thymectomy and add-on immunosuppressants might stabilize the clinical course. Therefore, prospective, and longitudinal follow-up of clinical assessment and miRNA levels might help to elucidate the value of prognostic biomarker of miRNA for MG flares.
Minor criticisms:
- Please provide the follow-up duration of the MG cohort.
- There are some typing errors needs for revision. For example, the numbers of participants should be 292 not 291 (17+10+245=20).「A total of 291 patients, whose serum samples passed quality controls, were 199 included in the final expression analyses: 17 LOMG, 10 EOMG, 245 HCs, 20 ONDs. (page 5, line 199-200)」
Author Response
Reviewer 2
- In this manuscript, the authors evaluated the serum levels of miRNAs in 27 MG patients, 245 healthy individuals and 20 patients with non-MG neuroimmune diseases and found high levels of miR-30e-5p predicted MG relapse (p=0.049) with the greatest potential as a predictive biomarker for the disease course in MG, regardless of subgroup. The paper should not be published in current context because of the criticisms as follows
Major criticisms:
- miRNAs have emerged as essential post-transcriptional regulators of gene expression and been evaluated as disease biomarkers for recent 2 decades. However, there were lots of confounding factors interfering the expression of miRNA including immunosuppressive agents. Eight out of 27 MG patients were NOT immunosuppressive naive, which might directly disturb the homogeneity of their MG cohort. Re-analysis using 19 immunosuppressive naive cohort might provide more solid base of comparison.
Response: We did not find a significant difference when comparing the two MG subgroups based on immunosuppression regarding any miRNA and believe we can include all patients on this basis. We expanded this in the Results section on page 6 (lines 251-255) and in the discussion on page 10 (lines 377-380).
- Clinical flares, the key event of disease course in this study, were defined as a change in MGC of > 3 points or change from a lower to a higher MGFA class and were documented in 15 out of 27 MG patients. Nerveless, the authors claimed the MGC scores were not available for all patients, and MGFA classes were not an optimal end point for clinical use. In addition, the treatment modalities including thymectomy and add-on immunosuppressants might stabilize the clinical course. Therefore, prospective, and longitudinal follow-up of clinical assessment and miRNA levels might help to elucidate the value of prognostic biomarker of miRNA for MG flares.
Response: We agree with this comment by the reviewer, and we have also outlined this as an important limitation of our study based on the in part retrospective design in the discussion on page 11 (lines 445-449). However, notwithstanding their flaws, our results suggest a predictive role for miR-30e-5p and this is in line with the previous prospective publication in regard to generalization from ocular MG (Sabre, L., et al., miR-30e-5p as predictor of generalization in ocular myasthenia gravis. Ann Clin Transl Neurol, 2019. 6(2): p. 243-251.) MG is a rare disease, and a single center prospective study is likely unfeasible. It is our hope that this publication may prove the foundation for larger international multicenter studies of prospective nature.
Minor criticisms:
- Please provide the follow-up duration of the MG cohort.
Response: We have added this information accordingly on page 5, lines 205-206.
- There are some typing errors needs for revision. For example, the numbers of participants should be 292 not 291 (17+10+245=20).「A total of 291 patients, whose serum samples passed quality controls, were 199 included in the final expression analyses: 17 LOMG, 10 EOMG, 245 HCs, 20 ONDs. (page 5, line 199-200)」
Response: We thank the reviewer for pointing this out and we have corrected this accordingly.

Round 2
Reviewer 1 Report
No further comments
Reviewer 2 Report
Re-analysis of MG cohort by immunosuppressive status had been done and the results were confident. The paper could be published in current context.